# Evaluation of a community-based performance arts programme for people who have experienced stroke in the UK: protocol for the SHAPER-Stroke Odysseys study

Carolina Estevao ,[1] Maria Baldellou Lopez,[2] Rachel E Davis ,[2]
Lucinda Jarret,[3] Tayana Soukup ,[2,4] Ioannis Bakolis,[2,5] Andy Healey,[2,6]
Jean Harrington,[1,7] Anthony Woods,[1,7] Nikki Crane,[8] Fiona Jones,[9]
Carmine Pariante,[1,10] Daisy Fancourt ,[11] Nick Sevdalis[2,8]

CE and MBL are joint first authors.

For numbered affiliations see end of article.

**Correspondence to**
Dr Carolina Estevao;
carolina.estevao@kcl.ac.uk

## ABSTRACT

**Introduction** Stroke survivors, once in the community, face challenges with their long-term rehabilitation care and present higher levels of loneliness, depression and anxiety than the rest of the population. A community-based performance arts programme, Stroke Odysseys (SO), has been devised to tackle the challenges of living with stroke in the UK. In this study, we aim to evaluate the implementation, impact and experiences of SO for stroke survivors.

**Methods and analysis** Scaling-up Health Arts Programmes: Implementation and Effectiveness Research (SHAPER)-SO aims to scale-up SO to 75 participants and 47 stakeholders, while simultaneously evaluating the effectiveness and implementation of the programme. The main research aim is to evaluate the implementation, effectiveness, impact and experiences of a community-based performance arts programme (SO for stroke survivors). This mixed-methods study will evaluate the experience and impact of SO on those participating using mixed methods (interviews, observations and surveys) before and after each stage and carry out non-participant observations during a percentage of the workshops, training and tour. Data will be analysed using quantitative and qualitative approaches. This is a study within the SHAPER programme.

**Ethics and dissemination** Ethical approval has been granted by the King's College London PNM Research Ethics Panel, REC reference: LRS/DP-20/21–21549. Written informed consent will be sought for participants and stakeholders. The results of the study will be reported and disseminated at international conferences and in peer-reviewed scientific journals.

**Trial registration number** NCT04864470.

## INTRODUCTION

Stroke affects over 113 000 people every year[1] and, according to the latest statistics, there are currently more than 1.2 million stroke survivors in the UK.[2 3] The effects of stroke are often devastating, with almost two-thirds of survivors leaving the hospital with a disability and half experiencing depression within 5 years.[4 5] In addition to the substantial impact, stroke has on those affected and their caregivers, it can also pose a significant financial burden to health and social care services. The societal cost of stroke has been estimated to be £26 billion per annum, with National Health Service (NHS) costs accounting for £3.4 billion in 2015, and projected to increase to £10.2 billion by 2035.[6]

Stroke survivors commonly face emotional, social and psychological challenges, with depression, anxiety and apathy being the most prevalent neuropsychiatric sequelae.[7] Such disabling symptoms are often coupled with feelings of abandonment[8] once hospital rehabilitation ends and their recovery plateaus. Stroke survivors in the UK usually receive rehabilitation while in hospital but once they

are discharged, the level of support in the community tends to be variable and in the long-term, inadequate for their needs.[8] This is consistent with a meta-review of qualitative systematic reviews,[9] which reported a lack of self-management resources available following stroke, highlighting the gap between available services and the long-term social, emotional and physical needs of stroke survivors throughout their rehabilitation journey.[10] Additionally, the findings of a survey by the Stroke Association in the UK emphasised the devastating burden and 'hidden effects' of stroke.[11] The survey, which collated data from over 10 000 stroke survivors and is the biggest to date in the UK, revealed that the effects of stroke on cognition, emotions, relationships and mental health are widespread, can be life-long and are often overlooked or neglected. In the survey, 50% of stroke survivors and 85% of caregivers reported a gap between the support provided versus the support they felt was needed. While current stroke rehabilitation targets functional recovery, it fails to meet the psychosocial needs of stroke survivors.

## Strengths and limitations of this study

► The first study examining an art intervention on stroke survivors, using a type 2 hybrid design, with a dual focus on effectiveness and implementation outcomes.

► The unique study design will result in a package of clinical and implementation data on this particular intervention.

► There may be inconsistency in participant experience throughout an intervention period if in-person sessions are switched online and vice versa due to COVID-19 social distancing restrictions.

► Access to the COVID-19-adapted online delivery of Stroke Odysseys may be challenging for people with severe acquired brain injury, resulting from stroke.

The evidence summarised above suggests that there is a need for more holistic rehabilitation programmes, especially non-pharmacological and non-invasive modalities, to address the psychosocial needs and improve the quality of life of stroke survivors.[12] Arts-based programmes (such as 'Stroke Odysseys (SO)' discussed below) are one such approach that shows promising results in enhancing the well-being, self-esteem, social life and rehabilitation experiences of patients with stroke.[13] Indeed, over the past decade, several studies conducted in this patient population have consistently shown a positive impact of different art modalities on psychological (eg, enhancement in confidence and a better sense of control), social (eg, increased social interactions and peer support) and functional (eg, improvement in physical abilities) outcomes.[12]

Nonetheless, despite the growing body of research on the benefits of art interventions, the process of scaling-up these interventions, embedding them into healthcare and its associated challenges are not yet well established. Preliminary data indicate that SO (discussed below) is received positively by those who take part,[14] however, identifying barriers to implementation and exploring ways to overcome these obstacles are essential to successfully and sustainably embed SO into clinical pathways and roll out the programme at a wider scale.

SO is part of the Scaling-up Health Arts Programme: Implementation and Effectiveness Research (SHAPER), which is, to our knowledge, the world's largest study on arts and health examining both clinical effectiveness and implementation effectiveness of three community-based arts programmes: Melodies for Mums (M4M), a singing intervention for postnatal depression, PD-Ballet, a dance intervention for Parkinson's Disease and SO. Overall, SHAPER has three primary aims: (1) to successfully embed each of the art interventions into the healthcare system (ie, taking a social prescribing approach), (2) to scale up these interventions at a larger scale and (3) to facilitate these interventions being commissioned by

Clinical Commissioning Groups (CCGs), ensuring the long-term sustainability of delivery.[15]

## Stroke Odysseys

Rosetta Life, a well-established non-profit organisation with a track record of conducting arts programmes for stroke and brain injury survivors, developed SO, a performance-based arts programme with continued consultation from stakeholders (including stroke survivors). SO provides an opportunity for those who have had a stroke or brain injury to share their experiences with an audience through movement, music, songwriting and the spoken word. The programme, which has now been running for over 21 years, uses performance arts to help stroke survivors overcome psychological challenges such as lowered self-esteem, anxiety and depression, which are commonly reported by individuals.[16]

In this protocol paper, we present our plans to evaluate. SHAPER-SO will be a two-pronged study, examining the implementation and clinical effectiveness of SO. The research we will be undertaking examines both, the impact of performance arts on participants and how SO can be embedded into clinical pathways. This will help us to identify not just 'if' but also 'why' the programme works and support our understanding of how it can be successfully delivered and scaled up within clinical pathways. Alongside this, we will examine participants' experiences of the programme using an ethnographic and constructivist approach. To the best of our knowledge, SHAPER-SO is the first study of its kind in the context of stroke care and rehabilitation.

## AIMS AND OBJECTIVES

The three main objectives in this study are: (1) to explore the clinical impact (effectiveness) of SO on stroke survivors; (2) to explore SO implementation aspects including uptake, adoption, perceived acceptability, appropriateness, feasibility, the fidelity of receipt, unintended consequences and sustainability and (3) to evaluate implementation costs and cost-effectiveness of the intervention, with focus on the costs associated with implementing SO into existing care pathways, health services, partner organisations and commissioning and the impact of scaling up SO on the utilisation of health services.

The main research aim is to evaluate the implementation, effectiveness, impact and experiences of a community-based performance arts programme (SO for stroke survivors).

Our study objectives are as follows:
1. To explore the impact of participation in performance programmes on cognitive health and physical, psychological and social well-being of people who have experienced stroke.
2. To study the context, mechanisms of delivery and interactions between participants and facilitators which take place during SO delivery.

3. To explore the learning and experiences of facilitators and participants after SO delivery.
4. To evaluate any change in the emotional well-being, participation and activity of stroke participants pre-SO and post-SO.
5. To evaluate the extent to which SO is acceptable, feasible to undertake and appropriate to survivors and wider stakeholders (including ambassadors, artists, and clinician referrers to the programme).
6. To explore the challenges, barriers, facilitators and unintended consequences of the implementation of SO.
7. To assess the costs associated with the implementation of the programme.
8. To assess the adoption, adherence to it and attrition rates of the programme.

## THEORETICAL UNDERPINNING

An ethnographic and constructivist approach will be used to examine stroke survivors' experiences of the SO programme (objective 2). This is described as the study of social interactions, behaviours and perceptions that occurs within groups, team organisation and communities. Ethnography provides rich, holistic insights into people's views and actions as well as the nature of the location (context) they inhabit. The aim has been described as 'getting inside' the way each group of people sees the world.[17] Ethnography has a strong emphasis on 'unstructured data and involves implicit interpretation of the meaning and function of human interactions, rather than hypothesis testing. This approach aligns well with the complex nature of the SO programme.

The implementation analyses are informed by several well-established implementation science frameworks, which we have applied to develop a set of implementation facets of SO to assess, both quantitatively and qualitatively (see the Methods). We used the recently developed 'Implementation Science Research Development' (ImpRes) framework[17] to identify the elements of implementation that the study ought to capture, ImpRes defined 10 different domains that an implementation evaluation ought to capture—including capturing stakeholder engagement, the outcome of implementation (eg, how acceptable, appropriate and feasible SO and its implementation processes are to those delivering and also receiving SO) and any unintended consequences (objective 3, 5 and 6). Moreover, we reviewed the Capability, Opportunity and Motivation Model of Behaviour (COM-B) tool[18] to help us identify any barriers that may affect an individual's engagement with the SO programme (objectives 7 and 10). The COM-B components lie at the centre of the Behaviour Change Wheel, a framework for designing and characterising behaviour change interventions.[18] The Consolidated Framework for Implementation Research (CFIR)[19 20] will help us map reported barriers and drivers to the implementation of the SO (objective 7); and finally, the Reach Effectiveness Adoption Implementation Maintenance model[21 22] taken together with Proctor et al's[23] taxonomy of implementation outcomes, guided our choice of implementation measures to assess.

## METHODS AND ANALYSIS

### Design

SHAPER-SO is a mixed-methods programme study, comprising quantitative and qualitative methods to assess the clinical and implementation outcomes outlined in the measures section below.

### Intervention

SO is a poststroke performance art intervention designed and delivered by the arts organisation Rosetta Life. This intervention initially developed and funded by King's and Guy's & St Thomas' Charity, aims to improve recovery, agency and well-being after stroke.[14]

SO comprises three distinct stages (1) weekly workshops conducted over 12 weeks for stroke participants which will be facilitated by an integrated team of expert artists and 'stroke ambassadors' from the charity Rosetta Life, (2) a smaller group of ambassadors recruited from the workshops will be trained to become cofacilitators (ie, new stroke ambassadors), (3) a performance tour including education and taster workshops for audiences.

During sessions, which run for 3 hours each, participants devise a dance and music performance work from their own stories. The practice of 'performing ourselves' is key to achieving successful outcomes such as transforming the participants' perception of identity. The culmination of the programme will be a public-facing performance to an audience of carers, healthcare practitioners, friends, family and the wider community.

Due to the ongoing COVID-19 pandemic and the necessity of shielding vulnerable adults and foreseeing increased anxiety in stroke survivors to attend in-person sessions, we have adapted the SO programme to be delivered through a mixture of live/face-to-face and online delivery (blended approach). Participants will be able to choose whether to attend the sessions/participate face-to-face or online based on their personal preferences and needs. The researcher will manage groups to ensure that all the participants who wish to attend in person will be able to do so during the 12 weeks.

The adapted programme will still be run in three stages, described below:

Stage (1): the workshops are the result of cocreation; the general framework is: weeks 1–3 building the performance company, weeks 4–6 devising the performance weeks 6–9 rehearsing the performance and weeks 10–12 are sometimes concertinaed into one production week introducing stage management, lighting and technical runs. Each of the 12 workshops contains a performance 'class' of 20–30 mins exploring movement and voice techniques and exercise.

Participants will be able to choose whether to attend the sessions/participate face-to-face or online based on their personal preferences and needs.

Stage (2): after the performance is completed, participants will be invited to a 4-day training programme where they will learn to act as advocates for life after stroke—termed 'stroke ambassadors'. The optional ambassador training starts with an introduction to being an ambassador and an outline of the pathways available: (a) supporting artists in hospital and community contexts, (b) speaking the press and media/advocating for life after stroke, (c) engaging in academic research and (d) joining the steering group that informs activities and directions.

The skill development training is delivered in three stages: an introduction to movement practices and the traditions of independent dance, then an introduction to voice and improvisation and, finally, an introduction to performance. Each ambassador then constructs an individually tailored programme according to their personal goals and intentions in becoming an ambassador.

The programme will take place once weekly and will be led by a team of artists and supported by a leadership coach. All training will take place on Zoom until social distancing measures are lifted, and participants are willing to meet indoors—a blended ambassador training will be offered.

Stage (3): following training, a volunteer manager will coordinate a tailored programme where ambassadors support artists in recruitment, befriend the newly discharged stroke survivors and take part in small-scale performance tours to challenge the perception of disability. The tour will be delivered online with online screenings followed by a question and answers (Q&A) session with ambassadors, taster sessions and exercises delivered online with the ambassadors.

The programme will be delivered in two cycles of the complete three-stage intervention. At the end of the two cycles of the programme, a group of newly trained ambassadors will emerge. The programme seeks to develop a national network of ambassadors who will build capacity for performance arts in healthcare and a wider capacity for healthcare. The stroke ambassadors are graduates of the 12-week workshop that receive training, based on a leadership-coaching model, and they deliver a tailored advocacy programme according to their creative skills—befriending, performance administration and support, programme advocacy.

## Study setting

The study will take place online until conditions of the pandemic enable researchers, artists and participants to meet safely indoors, as per government guidelines. When it is feasible and safe to meet in person, participants, artists and researchers will meet in an established performance arts education centre to ensure that COVID-19 guidelines on cleanliness are guaranteed. When ran in person, the workshops are run in Central London locations, with a single centre running the programme in each cycle.

## Sample and recruitment
### Stroke survivors

Consenting stroke participants will be included if they are:
1. over 18 years of age,
2. have had one or more stroke(s),
3. received inpatient care in a UK stroke care pathway,
4. able to follow a two-stage command and hold a conversation in English if no supporter/friend is available to translate.

The following exclusion criteria will be applied to individuals:
1. with comorbidities that would prevent participation in group activities (eg, dementia or deteriorating or fluctuating palliative conditions),
2. unable to understand English,
3. unable to commit to the 12-week programme.

Additionally, stroke ambassadors will be included if they have been through the ambassador training and are involved in at least one programme cycle culminating in the tour.

All participants will be offered the option of completing an interview with their carer present. This will be offered both after the first 12-week programme and after the ambassador training, for those that wish to participate. Those that decline will be asked if they would be willing to provide their reasons why.

### Wider stakeholder group

In addition to the stroke survivors that enrol on the SO programme, data will also be collected from a wider stakeholder group involved in the delivery or support of the programme. Individuals will be recruited if they meet the following criteria:
► over 18 years of age,
► can hold a conversation in English if no supporter/friend is available to translate,
► can either be defined as a:
  – Supporters: family members or carers.
  – Deliverers: individuals responsible for the delivery of the research (facilitators and artists).
  – Referrers: individuals involved in signposting (eg, doctors, nurses, healthcare workers).

Wider stakeholders will be excluded from participation if they are unable to understand English or if no supporter/friend is available to translate.

## Sampling
### Sample size

We aim to recruit 75 new stroke survivors in total for the duration of the study. A prediction of 75 participants has been estimated based on the numbers that over the years running SO, *Rosetta Life* has been able to recruit in two consecutive cycles. This number has also considered the organisation being able to while maintain a manageable ratio of participants to artists and staff members, guaranteeing that SO is delivered to the highest standard.

Based on previous experience of running SO where participants then complete an ambassador training cycle, a drop-out rate of 20% is expected, and so the final number of ambassadors that complete the ambassador training is estimated to be 60.

The wider stakeholder group will be recruited from the network of people who are involved in the programme and present in the community. This includes the voluntary sector, health and social care sectors and clinical commissioners. A total of 47 stakeholders, a forecast based on the existing network numbers, will be recruited (12 carers, 10 clinical team members, 5 artists, 20 existing ambassadors).

## Recruitment procedure

Potential stroke survivor participants will be identified through signposting in community centres and care homes as well as engaging in presentations, screenings, taster sessions and performances during the tour.

Recruitment of potential participants will be done online. Screenings of performance extracts will be followed by taster sessions online and a Q&A with ambassadors. Potential participants will be directed to the project manager at Rosetta Life who will manage all referrals.

Potential participants will be offered a Participant Information Sheet (PIS) and an Informed Consent Form (ICF) and will be explained the details of the study. Written consent will be sought following a 48 hours colling-off period.

Wider stakeholders will be recruited from the networks of people involved in the referral, delivery or support of the programme.

A recruitment log will be kept by the research team to accurately record included and excluded participants as well as missing data from dropouts to account for possible sampling bias.

## Study flowchart

A study overview is seen in the flowchart below (figure 1):

## Data collection

This is a prospective mixed-methods study using a range of qualitative and quantitative methods at different time points pre, during and postintervention of each programme cycle.

Qualitative methods will comprise semistructured interviews and non-participant observations of training and production to assess experiences and attitudes towards the programme and its implementation.

Quantitative methods will be used to assess experiences and attitudes towards the SO programme and its implementation. Further information is included further in the 'methods' section in the outcome measures tables (tables 1 and 2) and the 'assessment descriptions' section.

Demographic data will be collected by Rosetta Life at the time of enrolment.

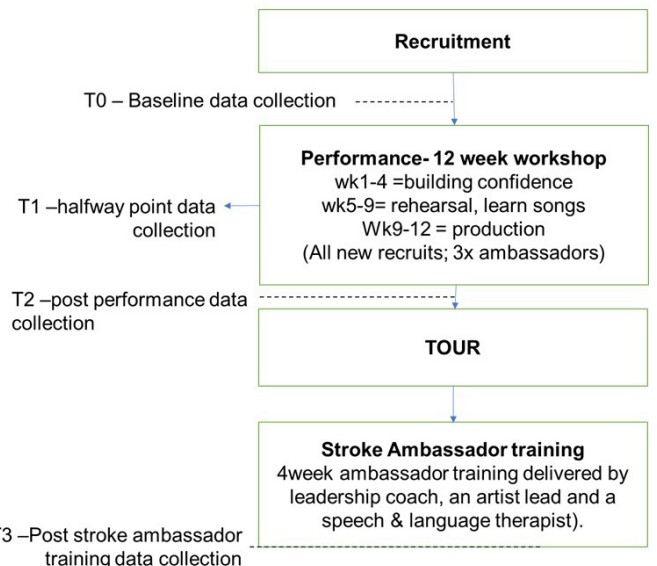

**Figure 1** SHAPER-SO study flowchart. Scaling-up Health Arts Programmes: Implementation and Effectiveness Research-Stroke Odysseys.

## Outcomes

Data on the clinical outcomes will be collected from stroke survivors who have enrolled on the SO programme (see table 1). Data on the implementation outcomes will be collected from stroke survivors who have enrolled on the SO programme as well as the wider stakeholder group involved in the SO programme, including deliverers, referrers and supporters (see table 2).

Time points for data collection: T0—baseline; T1—midway through the 12-week programme (weeks 5–7); T2—immediately postperformance (12–14 weeks); T3—immediately after the advocacy training for stroke ambassadors.

To maximise inclusivity and outcome completion, and minimise participant burden, outcome assessments, where possible, will be conducted either face-to-face, online, by telephone or via postal questionnaire depending on the outcome measures being assessed, participants' preferences and government COVID-19 guidelines.

### Assessment descriptions for clinical outcomes
#### Qualitative assessments
##### Ethnographic research

Ethnographic non-participant observations of a selection of the 12 workshops including at least 1–2 groups from each of the two phases (building confidence, rehearsal and production) to capture facilitator and participant practice, interactions and routines. Each observation period will last for the duration of the workshop, and the ethnographic researcher will record field notes contemporaneously.

##### Semistructured interviews

Semistructured interviews will be held with facilitators and participants' pre and postprogramme cycles to explore anticipated concerns and expectations (pre)

**Table 1** Clinical outcomes

| Objective | Clinical outcome measures/ endpoints | Type of assessment | The time point for data collection |
|---|---|---|---|
| **Primary objective** | | | |
| **Secondary objectives** | | | |
| To study the context, mechanisms and interactions which take place during SO delivery | Non-participant observations of workshops | Qualitative | T1 (during workshop delivery) |
| To explore the learning and experiences of facilitators and participants | Semi-structured interviews- stroke participants and facilitators | Qualitative | T2 |
| To explore stroke survivors' preparation and participation in performances | Semi-structured interviews- stroke participants | Qualitative | T0 |

Data on the clinical outcomes will be collected from stroke survivors who have enrolled on the SO programme.
SO, Stroke Odysseys.

and experiences of facilitation and factors influencing delivery, engagement of participants, adaptation and learning (post).

The implementation science research team will be interviewing participants across both, the 12-week programme and ambassador training, in addition to wider stakeholders.

### Assessment descriptions for the implementation outcomes
#### Quantitative assessments
Validated and standardised implementation scales will be used to gather quantitative data on how acceptable, appropriate and feasible the SO programme is perceived by stroke survivors, ambassadors, deliverers, supporters and referrers. These scales include the Acceptability of Programme Measure (AIM), the Programme Appropriateness Measure (IAM) and the Feasibility of Programme Measure (FIM). For further information on the development of these scales, please refer to the paper by Weiner *et al.*[24]

The implementation science researchers will quantify and cost the resources used in implementing the programme, evaluate wider service utilisation and associated costs before and after participants complete the programme, including any changes to their quality-of-life profile measured using the EQ5D-3L preference-based QoL measure. The EQ5D-3L is a self-complete multiattribute measure of health-related quality of life that assigns individuals a unique state of health based on their response to individual items. Each unique health state is associated with a predetermined 'utility' value derived from a survey of wider community preferences over different states of health. The utility scale is anchored at 1 (full health) and zero (death), with negative values allowed in instances where states of health are considered worse than death. Health state utility values are subsequently used to estimate quality-adjusted life years (QALYs) survived over time—the utility scores providing the means of making the quality adjustments. Evidence on costs and QALYs will subsequently be used to inform an analysis of the cost-effectiveness of programme delivery at scale.

### Qualitative data collection
#### Semistructured interviews
Semistructured interviews will be conducted with a purposive subsample of stroke survivors (N=20: 5 from each cycle at two time points—T2 and T3). Interviews will be carried out with this subsample of stroke survivors to explore their attitudes towards the acceptability, appropriateness and feasibility of the programme as well as factors (facilitators or barriers) that affected their involvement (and potential drop-out) and any unintended consequences. These issues will also be explored with a subsample of individuals (10 in total) from each of the wider stakeholder groups.

Interview guides have been based on the existing implementation frameworks (see above) and adapted from a previous project.[25] They will be further adapted and codesigned with our stakeholder group to ensure the questions in the interview guide are meaningful and address the core aims of the study.

Interviews will be audiotaped and are anticipated to be conducted 1:1 or in participants dyads, face to face (government guidelines permitting) or remotely by phone or video.

### Data analysis
Data will be analysed using quantitative and qualitative approaches.

### Quantitative analysis
Descriptive statistics of survey data will be performed (frequency distribution, central tendency). Parametric and non-parametric tests will also be employed to compare the survey responses to the AIM, FIM, IAM and EQ5D before and after the SO intervention. Changes in AIM, FIM, IAM and EQ5D will be assessed using generalised linear models depending on the distribution of the

**Table 2** Implementation outcomes

| Objective | Implementation outcome measures/endpoints | Type of assessment | Time points for data collection | Who data will be collected from |
|---|---|---|---|---|
| **Primary objective** | | | | |
| To evaluate to what extent SO is acceptable, to survivors and wider stakeholders | Acceptability of intervention Measure Semi-structured interviews (to explore reasons for acceptability score) | Quantitative Qualitative | T1, T2, T3 T2, T3 | Stroke survivors, deliverers, supporters, referrers Stroke survivors, deliverers, supporters, referrers |
| **Secondary objectives** | | | | |
| To evaluate to what extent SO are appropriate to survivors and wider stakeholders | Intervention Appropriateness Measure Semi-structured interviews (to explore reasons for appropriateness score) | Quantitative Qualitative | T1, T2, T3 T2, T3 | Stroke survivors, deliverers, supporters, referrers Stroke survivors, deliverers, supporters, referrers |
| To evaluate to what extent Stroke Odysseys feasible to survivors and wider stakeholders | Feasibility Intervention Measure Semi-structured interviews (to explore reasons for feasibility score) | Quantitative Qualitative | T1, T2, T3 T2, T3 | Stroke survivors, deliverers, supporters, referrers Stroke survivors, deliverers, supporters, referrers |
| To assess any unintended consequences of the programme | Semi-structured interviews | Qualitative | T2, T3 | Stroke survivors, deliverers, supporters, referrers |
| To explore the facilitators and barriers to implementing the programme | Semi-structured interviews | Qualitative | T2, T3 | Stroke survivors, deliverers, supporters, referrers |
| To explore the facilitators and barriers to sustained use of the programme | Semi-structured interviews | Qualitative | T2, T3 | Stroke survivors, deliverers, supporters, referrers |
| To assess service utilisation and cost associated costs and changes in quality of life associated with the implementation of the programme | EQ5D-5L (quality of life measure) and AD-SUS (adult service receipt schedule) and semi-structured interviews and activity data (to estimate implementation costs). | Quantitative | T2 and T3 | Stroke survivors Stroke survivors, deliverers, supporters, referrers |
| To explore the strategies including resource inputs used, used within individual sites to implement the programme | Semi-structured interviews | Qualitative | T2, T3 | Deliverers, referrers |
| To assess the adoption of the programme | The number of individuals delivering the programme, and the number of individuals supporting the programme (and continuing to do so) | Quantitative | T0, T2, T3 | Deliverers, referrers |
| To assess programme adherence and attrition rates | Data on the overall adherence to the programme, number of drops-outs and reasons why | Quantitative Qualitative | Data recorded from the register on weekly attendance rates for the 12 week programme (stage 1) and 4 week ambassador programme (stage 2) T2, T3 | Deliverers (record data) Stroke survivors |

Data on the implementation outcomes will be collected from stroke survivors who have enrolled on the SO programme as well as the wider stakeholder group involved in the SO programme (including deliverers, referrers and supports).
[1] Adult Service Use Schedule (AD-SUS)
SO, Stroke Odysseys.

outcome (continuous, binary, ordinal). All analyses will be conducted in STATA V.14.1.

### Qualitative analysis

Initial analysis of qualitative data will be undertaken using an inductive approach to thematic analysis. All data from interviews and observations will be managed using NVivo V.10 and examined to categorise themes and key issues that emerge. Using this inductive approach, tentative theoretical explanations will be generated for each subgroup. Summary memos for data sets will be developed for each subgroup to provide the basis for within and between-group comparisons. The inductive approach is data driven; based on observation, the early analysis seeks to reveal patterns and themes from which tentative hypothesis can be drawn subsequently leading to theory; theories are devised to explain what is seen rather than the other way around.

CFIR (www.CFIR.org) will be used to further guide the coding and analysis (ie, framework analysis) of interview data to identify barriers and facilitators to the implementation and sustainment of the SO programme. This approach has been used previously, that is, CFIR has been

applied postimplementation to investigate facilitators and barriers to implementation among stakeholders who had already adopted and implemented an innovation, thus identifying determinants of implementation posthoc.[26 27]

Reflective summaries: the relationship of the researcher(s) with the research context they are investigating will be presented in the form of a written narrative of ideas and experiences during data collection. These reflective summaries will be shared with the research team and externally to judge any possible biases with the way the data were collected or prior assumptions.

### Patient and public involvement

The programme has been developed and further refined using codesign methodologies with a group of 20 members of South London stroke communities. The project has been shared widely with stroke clinicians across London and has their full support. During the pandemic Rosetta Life set up an advisory group consisting of Stroke Ambassadors to support the redesign of the website www.strokeodysseys.org, to monitor how people living with the effects of a stroke were engaging with the online workshops, to oversee the development of the education videos and the Ambassadors Handbook.

This advisory group is now a stable and national network of ambassadors who curate an online programme and advise on the development and delivery of SO. They have advised the investigators on the need to ensure that the measures were aphasia friendly and found an organisation to make sure that the measures were aphasia friendly. They will now look at the language of the Implementation Science measures and make sure that they are accessible.

### Trial registration and current status

This study is registered on ClinicalTrials.gov PRS under the ClinicalTrials.gov. Recruitment was scheduled to start in Autumn 2021.

### Data protection

The investigator will ensure that this study is conducted in full conformity with relevant regulations and with the ICH Guidelines for Good Clinical Practice (CPMP/ICH/135/95) July 1996. The investigator will ensure that this study is conducted in accordance with the principles of the Declaration of Helsinki.

Access to person identifiable implementation science data will rest with the data custodian(s) from the immediate study team and the implementation science team. Since the project seeks to explore in some depth participants' experiences and barriers and facilitators to implementation, it is important to maintain strict confidentiality and facilitate openness in the interviews and survey responses, thus optimal data quality.

Consent forms and audio/video recordings will be kept electronically in KCL's SharePoint for the duration of the study, only accessible by the teams at KCL, Kingston University and Rosetta Life involved in the study. Consent forms and other identifiable paperwork will be kept in locked cabinets only accessible to the study team. Study data will be kept in a separate location from the person identifiable information. Access to the deidentified research data will be shared with the study management group for the purposes of review, analysis and dissemination. Only deidentified data will be analysed.

After the completion of the study, the study data will be kept for the King's College London's standard retention period of 10 years after the completion of the study. The study data that support published results will be deposited in a secure data repository (eg, King's Research Data Management System). This will allow the data to be accessible for future reuse as per King's College London's policy on the management of research data long-term.

## ETHICS AND DISSEMINATION

Ethical approval has been granted by the King's College London PNM Research Ethics Panel, REC reference: LRS/DP-20/21–21549. Informed consent will be collected in writing from all research participants and stakeholders involved in the study. Findings will be published in peer-reviewed journals and disseminated at national and international meetings.

**Author affiliations**
[1]Psychological Medicine, Institute of Psychiatry, Psychology & Neuroscience, King's College London, London, UK
[2]Centre for Implementation Science, King's College London Institute of Psychiatry Psychology and Neuroscience, London, UK
[3]Rosetta Life Head Office, London, UK
[4]Centre for Implementation Science, King's College London, London, UK
[5]Department of Biostatistics and Health Informatics, Institute of Psychiatry, Psychology & Neuroscience, London, UK
[6]Health Service and Population Research Department, King's Health Economics, King's College London, London, UK
[7]Department of Psychological Medicine, King's College London, London, UK
[8]Culture Team, King's College London, London, UK
[9]Faculty of Health, Social Care and Education, Kingston University and St George's, University of London, London, UK
[10]Psychological Medicine, King's College London, London, UK
[11]Department of Behavioural Science and Health, University College London, London, UK

**Contributors** All authors listed have contributed to the conception and design of the protocol and this manuscript. All authors have been involved in the drafting of the manuscript and have individually approved the version of the work published. Specifically, the contribution of each author falls within the following CRediT categories: CE, MBL, RED, TS, IB, JH, AH, FJ and DF, NS: conceptualisation, methodology and project administration. AW, NC, CP: conceptualisation, project administration and funding acquisition. LJ: conceptualisation, project administration.

**Funding** This research was funded in whole by the Wellcome Trust [219425/Z/19/Z]. For the purpose of open access, the author has applied a CC BY public copyright licence to any Author Accepted Manuscript version arising from this submission.This trial is part of the SHAPER programme, a Scaling-up Health-Arts Programme to scale up arts programmes. This work is additionally supported by the National Institute for Health Research (NIHR) Biomedical Research Centre at South London and Maudsley NHS Foundation Trust and King's College London and by an NIHR Senior Investigator to CMP.

**Competing interests** NS is the director of London Safety and Training Solutions Ltd, which offers training in patient safety, implementation solutions and human factors to healthcare organisations and the pharmaceutical industry. CP reports

grants from the Wellcome Trust, during the conduct of the study; and grants from the National Institute for Health Research (NIHR), NIHR Senior Investigator, Johnson & Johnson, and the Wellcome Trust, outside the submitted work. The other authors have no conflicts of interest to declare.

**Patient and public involvement**  Patients and/or the public were involved in the design, or conduct, or reporting, or dissemination plans of this research. Refer to the Methods section for further details.

**Patient consent for publication**  Not applicable.

**Ethics approval**  Ethical approval has been granted by the King's College London PNM Research Ethics Panel, REC reference: LRS/DP-20/21-21549. Written informed consent will be sought for participants and stakeholders. The results of the study will be reported and disseminated at international conferences and in peer-reviewed scientific journals.

**Provenance and peer review**  Not commissioned; externally peer reviewed.

**ORCID iDs**
Carolina Estevao http://orcid.org/0000-0001-7758-0371
Rachel E Davis http://orcid.org/0000-0003-2406-7181
Tayana Soukup http://orcid.org/0000-0003-0203-7264
Daisy Fancourt http://orcid.org/0000-0002-6952-334X

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
