## [Reviewer comments · BMJ Open]

ARTICLE DETAILS

TITLE (PROVISIONAL)	Evaluation of a community-based performance arts programme for people who have experienced stroke in the UK: protocol for the SHAPER-Stroke Odysseys study
AUTHORS	Estevao, Carolina; Baldellou Lopez, Maria; Davis, Rachel; Jarret, Lucinda; Soukup, Tayana; Bakolis, Ioannis; Healey, Andy; Harrington, Jean; Woods, Anthony; Crane, Nikki; Jones, Fiona; Pariante, Carmine; Fancourt, Daisy; Sevdalis, Nick

VERSION 1 – REVIEW

REVIEWER	Sureshkumar, K Public Health Foundation of India, SACDIR
REVIEW RETURNED	24-Oct-2021

GENERAL COMMENTS	It is an excellent concept and I congratulate the authors for their landmark study in the UK. I am enclosing my comments and I believe that this paper will be very useful for the readers if it's revised substantially. "The reviewer provided a marked copy with additional comments. Please contact the publisher for full details."
--

REVIEWER	Markovic, Gabriela Karolinska Institute
REVIEW RETURNED	22-Nov-2021

GENERAL COMMENTS	Thank you for a wellwritten protocoll and interesting study on possible outcome from a community-based programme of performance arts for stroke survivors. It is indeed challenging for our patients once they leave the rehabilitation centre and patient organisations can only comply for some of the issues. It is also very exciting that you have focused on a qualitative approach, as I believe it is indeed the proper method for exploring these questions. From my own experiences in running research projects, it is all clear in my head and when I talk about the project I truly believe everyone knows what I am talking about. With that little anecdotal intro of mine, I think your study protocol would benefit from an upgraded clarification. It is quite useful, as you can refer to the protocol in all coming manuscripts with no need to repeat info. It is really only a question of clarifications of the rationale and the intervention per se. I wish I was onboard. Looking forward to an updated protocol! Introduction
---

	You bring forth compelling arguments for the challenges stroke survivors face once hospital rehabilitation ends. The introduction would benefit from equally compelling arguments that art-based programmes could meet those needs. Are there other programmes for example, that have proven beneficial? On line 45 and 53 (page 4) respectively you outline depression and anxiety specifically. Is there a reason for that? Reduced levels of depression and anxiety are at most a secondary goal to the intervention (or a beneficial consequence?). The questions referring to mental well-being included in the Ox-PAQ are indicators of current mental health status, and do not provide information on level of depression or anxiety (mental illnesses). Maybe you could, in the study protocol, omit specificities about depression and anxiety as they are not part of the study objective. Maybe add an argument or two advocating for the need of studying the implementation process. It would help the reader understand the rationale behind the aims and objectives. Aims and objectives Do I understand it correctly that the main aim is to evaluate implementation and explore experiences of the programme? From the introduction I got the idea that it was about the participants experience and possible benefit of the programme at large, and possibly on a secondary note (and pre-supposing a positive outcome) an exploration of the implementation process, facilitators, barriers, and costs of such. The research aims and study objectives are clear, and well defined. Unfortunately, the introduction doesn't really show me the importance behind the questions. A few clarifications of the study objectives are demanded:  (1) Study objective 1: Explore the impact of participation in the programme on what? On mental health, participation, self-esteem? (line 57-58; p 5) (2) Study objective 3: complete the sentence (line 6; p 6). (3) Study objective 6 (line 13; p 6): raises the question of 'intended' consequences which are not pronounced elsewhere in the protocol. Is it a question of adverse effects? Methods and analysis The intervention  • Rosetta Life, and Stroke Odysseys, are well described on the homepage. The work they do is simply beautiful. However, I do believe the study protocol would benefit from an expanded intervention
--	---

	description and rationale. As a reader of the protocol, I have yet to understand what it is the stroke survivors will do, the factual performance - what counts as/is included in performance arts. Please provide a more detailed description, either in text or in a table.  • describe the contents of ambassador training - what is included. Is becoming ambassador one of the stages of the programme? • How often are these programmes conducted, bi-annually, tertially? Study setting  • Add a line describing how many centres are involved, and that a description of these centres will follow in future studies. Are the centres in urban or rural areas? Sample and Recruitment  • will all participants completing the programme be offered ambassador training, or are they 'hand-picked'? Please provide additional information of the recruitment process, including exclusion and inclusion criteria. • Out of curiosity, what is the aim and purpose of ambassadors? Please clarify the rationale behind having ambassadors and include if they are oriented towards programme participating only or if they area also ambassadors for stroke survivors in other organisations such as hospitals, patient organisations. • What happens to those who decline ambassador training? Will they too (including wider stakeholders if family members or carers) be subject to assessment and interviews? Sampling  • Size of sample  ○ Rationale for the 75 and 47 included. Is it for power-purposes of the statistical instrument? Please provide power calculation for this. ○ Power is obviously not needed for studies of qualitative design, but please provide an estimation of number of interviews needed. • Recruitment procedure  ○ Describe who will be managing the recruitment process, that is, describe
--	---

	who keeps track of the recruitment log for you to present an elegant inclusion flowchart in future papers, accounting for possible sampling bias.  ○ As this study is an intervention study with ethical approval, describe the whereabouts of informed and written consent. Potential stroke survivors will see advertisements through signposting and be approached during tours. Will there be information also about the study and not only about the programme? Those who do not wish to participate, are they still enrolled in the programme? To clarify, I am not asking for programme adherence which is well described as an objective in Table 1, but for compliance to study participation. Data collection  • Outcomes  ○ Describe who does the assessments and who will be conducting interviews ○ Describe the process around data storage • Assessment descriptions (page 13)  ○ Ethnographic research (line 20): I read '12 workshops' and get a bit confused. Previously you mention a 12-week programme with workshops, are they in fact 12 workshops 1 week each? If so, please include a table shortly describing the 12 workshops. ○ Quantitative assessments (line 58-60): 'investigators will quantify and cost the resources used' – it might be my level of English understanding, but I do not comprehend what they will do. Will the evaluation of the costs (direct and associated) be done in relation to changes in QoL as measured by the Eq5D? ○ Provide a short description of the Eq5D, including how you plan to deal with the visual analogue scale, EQ VAS given that some assessment may be done online. • Qualitative data collection (page 14)
--	--

	 ○ Lines 14-15: clarify whether you refer to the sub-sample when you mention '10 in total' or to number of wider stakeholder groups. If the latter, it ought to be outlined elsewhere in the method section. Also, please provide information about the following  • demographic and clinical data included. If not included, please provide rationale for that. • Number of groups are needed to reach patient recruitment • Size of group, including wider stakeholders • time for data collection, i.e., when does the study start and expected end of data collection (approximate date) Data analysis (page 14)  • Qualitative analyses (line 42): you will be using an inductive thematic approach. Please clarify ad modern what template. It would provide the reader enough information about the theoretical framework for the analytical method chosen. I apologize if you think me too meticulous, but I do believe the clarifications are somewhat necessary. I want to emphasize the importance of study protocols both for the research community and the research group as they outline the what, the why and the how of larger data collections. Looking forward to a revised manuscript.
--	---

VERSION 1 – AUTHOR RESPONSE

Introduction

You bring forth compelling arguments for the challenges stroke survivors face once hospital rehabilitation ends. The introduction would benefit from equally compelling arguments that art-based programmes could meet those needs. Are there other programmes for example, that have proven beneficial?

On line 45 and 53 (page 4) respectively you outline depression and anxiety specifically. Is there a reason for that? Reduced levels of depression and anxiety are at most a secondary goal to the intervention (or a beneficial consequence?). The questions referring to mental well-being included in the Ox-PAQ are indicators of current mental health status, and do not provide information on level of depression or anxiety (mental illnesses). Maybe you could, in the study protocol, omit specificities about depression and anxiety as they are not part of the study objective.

Maybe add an argument or two advocating for the need of studying the implementation process. It would help the reader understand the rationale behind the aims and objectives.

Aims and objectives

Do I understand it correctly that the main aim is to evaluate implementation and explore experiences of the programme? From the introduction I got the idea that it was about the participants experience and possible benefit of the programme at large, and possibly on a secondary note (and pre-supposing a positive outcome) an exploration of the implementation process, facilitators, barriers, and costs of such. The research aims and study objectives are clear, and well defined. Unfortunately, the introduction doesn't really show me the importance behind the questions.

A few clarifications of the study objectives are demanded:

- (1) Study objective 1: Explore the impact of participation in the programme on what? On mental health, participation, self-esteem? (line 57-58; p 5)
- (2) Study objective 3: complete the sentence (line 6; p 6).
- (3) Study objective 6 (line 13; p 6): raises the question of 'intended' consequences which are not pronounced elsewhere in the protocol. Is it a question of adverse effects?

Methods and analysis

The intervention

- Rosetta Life, and Stroke Odysseys, are well described on the homepage. The work they do is simply beautiful. However, I do believe the study protocol would benefit from an expanded intervention description and rationale. As a reader of the protocol, I have yet to understand what it is the stroke survivors will do, the factual performance - what counts as/is included in performance arts. Please provide a more detailed description, either in text or in a table.
- describe the contents of ambassador training - what is included. Is becoming ambassador one of the stages of the programme?
- How often are these programmes conducted, bi-annually, tertially?

Study setting

- Add a line describing how many centres are involved, and that a description of these centres will follow in future studies. Are the centres in urban or rural areas?

Sample and Recruitment

- will all participants completing the programme be offered ambassador training, or are they 'hand-picked'? Please provide additional information of the recruitment process, including exclusion and inclusion criteria.
- Out of curiosity, what is the aim and purpose of ambassadors? Please clarify the rationale behind having ambassadors and include if they are oriented towards programme participating only or if they area also ambassadors for stroke survivors in other organisations such as hospitals, patient organisations.
- What happens to those who decline ambassador training? Will they too (including wider stakeholders if family members or carers) be subject to assessment and interviews?

Sampling

- Size of sample
 - o Rationale for the 75 and 47 included. Is it for power-purposes of the statistical instrument? Please provide power calculation for this.
 - o Power is obviously not needed for studies of qualitative design, but please provide an estimation of number of interviews needed.
- Recruitment procedure
 - o Describe who will be managing the recruitment process, that is, describe who keeps track of the recruitment log for you to present an elegant inclusion flowchart in future papers, accounting for possible sampling bias.
 - o As this study is an intervention study with ethical approval, describe the whereabouts of informed and written consent. Potential stroke survivors will see advertisements through signposting and be approached during tours. Will there be information also about the study and not only about the programme? Those who do not wish to participate, are they still enrolled in the programme? To

clarify, I am not asking for programme adherence which is well described as an objective in Table 1, but for compliance to study participation.

Data collection

- Outcomes
 - o Describe who does the assessments and who will be conducting interviews
 - o Describe the process around data storage
- Assessment descriptions (page 13)
 - o Ethnographic research (line 20): I read '12 workshops' and get a bit confused. Previously you mention a 12-week programme with workshops, are they in fact 12 workshops 1 week each? If so, please include a table shortly describing the 12 workshops.
 - o Quantitative assessments (line 58-60): 'investigators will quantify and cost the resources used' – it might be my level of English understanding, but I do not comprehend what they will do. Will the evaluation of the costs (direct and associated) be done in relation to changes in QoL as measured by the Eq5D?
 - o Provide a short description of the Eq5D, including how you plan to deal with the visual analogue scale, EQ VAS given that some assessment may be done online.
- Qualitative data collection (page 14)
 - o Lines 14-15: clarify whether you refer to the sub-sample when you mention '10 in total' or to number of wider stakeholder groups. If the latter, it ought to be outlined elsewhere in the method section.

Also, please provide information about the following

- demographic and clinical data included. If not included, please provide rationale for that.
- Number of groups are needed to reach patient recruitment
- Size of group, including wider stakeholders
- time for data collection, i.e., when does the study start and expected end of data collection (approximate date)

Data analysis (page 14)

- Qualitative analyses (line 42): you will be using an inductive thematic approach. Please clarify ad modum what template. It would provide the reader enough information about the theoretical framework for the analytical method chosen.

I apologize if you think me too meticulous, but I do believe the clarifications are somewhat necessary. I want to emphasize the importance of study protocols both for the research community and the research group as they outline the what, the why and the how of larger data collections. Looking forward to a revised manuscript.

Be well,

Reviewer #1

Introduction

1. You bring forth compelling arguments for the challenges stroke survivors face once hospital rehabilitation ends. The introduction would benefit from equally compelling arguments that art-based programmes could meet those needs. Are there other programmes for example, that have proven beneficial?

Other similar programmes are beneficial for this population. Added to the introduction section: " Arts-based programmes (such as 'Stroke Odysseys' discussed below) are one such approach that shows promising results in enhancing the wellbeing, self-esteem, social life and rehabilitation experiences of

stroke patients [15]. Indeed, over the past decade, several studies conducted in this patient population have consistently shown a positive impact of different art modalities on psychological (e.g. enhancement in confidence and a better sense of control), social (e.g. increased social interactions and peer support) and functional (e.g. improvement in physical abilities) outcomes [14].”

2. On line 45 and 53 (page 4) respectively you outline depression and anxiety specifically. Is there a reason for that? Reduced levels of depression and anxiety are at most a secondary goal to the intervention (or a beneficial consequence?). The questions referring to mental well-being included in the Ox-PAQ are indicators of current mental health status, and do not provide information on level of depression or anxiety (mental illnesses). Maybe you could, in the study protocol, omit specificities about depression and anxiety as they are not part of the study objective.

We have evidence that supports the hypothesis that Stroke Odysseys may indirectly offer improvements in the mental health (anxiety and depression) of stroke survivors, through the build of community and increased self-efficacy. Hence, we have highlighted depression and anxiety in the abstract and introduction as some of the common challenges that this population experiences and will investigate any potential effects through our mixed-methods study.

3. Maybe add an argument or two advocating for the need of studying the implementation process. It would help the reader understand the rationale behind the aims and objectives.

Added the following paragraph to the Introduction section: “Indeed, over the past decade, several studies conducted in this patient population have consistently shown a positive impact of different art modalities on psychological (e.g. enhancement in confidence and a better sense of control), social (e.g. increased social interactions and peer support) and functional (e.g. improvement in physical abilities) outcomes [14]. Nonetheless, despite the growing body of research on the benefits of art interventions, the process of scaling-up these interventions, embedding them into healthcare and its associated challenges are not yet well-established. Preliminary data indicates that Stroke Odysseys (discussed below) is received positively by those who take part, however, identifying barriers to implementation and exploring ways to overcome these obstacles is essential to successfully and sustainably embed SO into clinical pathways and roll out the programme at a wider scale.”

Aims and objectives

4. Do I understand correctly that the main aim is to evaluate the implementation and explore experiences of the programme? From the introduction, I got the idea that it was about the participants experience and possible benefit of the programme at large, and possibly on a secondary note (and presupposing a positive outcome) an exploration of the implementation process, facilitators, barriers, and costs of such. The research aims and study objectives are clear and well defined. Unfortunately, the introduction doesn't show me the importance of the questions.

Added the following paragraph to the Introduction section: “Stroke Odysseys is part of the Scaling-up Health-Arts Programme: Implementation and Effectiveness Research (SHAPER), which is, to our knowledge, the world's largest study on arts and health examining both, clinical effectiveness and implementation effectiveness of three community-based arts programmes: Melodies for Mums (M4M), a singing intervention for postnatal depression, PD-Ballet, a dance intervention for Parkinson's Disease, and Stroke Odysseys (SO). Overall, SHAPER has three primary aims: (1) to successfully embed each of the art interventions into the healthcare system (i.e. taking a social prescribing approach), (2) to scale up these interventions at a larger scale, and (3) to facilitate these interventions being commissioned by CCGs, ensuring the long-term sustainability of delivery [17].”

A few clarifications of the study objectives are demanded:

1. Study objective 1: Explore the impact of participation in the programme on what? On mental health, participation, self-esteem? (line 57-58; p 5)

The first aim of the study is to explore the impact of participation on the areas of cognitive health and physical, psychological and social well-being as they become manifest through observation and within conversations. Reworded the first objective to: "To explore the impact of participation in performance programmes on cognitive health and physical, psychological and social well-being of people that have experienced stroke."

2. Study objective 3: complete the sentence (line 6; p 6).

Amended objective 3 to: "To explore learning and experiences of facilitators and participants after SO delivery."

3. Study objective 6 (line 13; p 6): raises the question of 'intended' consequences which are not pronounced elsewhere in the protocol. Is it a question of adverse effects?

By "unintended consequences" it is meant that there may be other consequences of the intervention to participants and stakeholders that are not foreseen at the protocol writing stage (i.e improvement in other social relationships, etc.), not adverse effects.

Methods and analysis

The intervention

4. Rosetta Life, and Stroke Odysseys, are well described on the homepage. The work they do is simply beautiful. However, I do believe the study protocol would benefit from an expanded intervention description and rationale. As a reader of the protocol, I have yet to understand what it is the stroke survivors will do, the factual performance - what counts as/is included in performance arts. Please provide a more detailed description, either in text or in a table.

A description of the intervention has been expanded in the section "The intervention". I hope this updated section offers the level of detail you wished.

5. Describe the contents of ambassador training - what is included. Is becoming an ambassador one of the stages of the programme?

Thank you for your comment. Becoming an ambassador is an option of the programme after the 12-week workshop. I have added to the description of the intervention (Stage 2): "The optional ambassador training starts with an introduction to being an ambassador and an outline of the pathways available: a) supporting artists in hospital and community contexts, b) speaking the press and media / advocating for life after stroke, c) engaging in academic research and d) joining the steering group that informs activities and directions.

The skills development training is delivered in three stages: an introduction to movement practices and the traditions of independent dance, then an introduction to voice and improvisation and finally, an introduction to performance. Each ambassador then constructs an individually tailored programme according to their personal goals and intentions in becoming an ambassador."

6. How often are these programmes conducted, bi-annually, tertially?

As mentioned throughout the protocol (intervention, sample, data collection sections), the programme will run in two cycles. Note from the lead artist: Ideally these programmes are run biannually but they are funding dependent.

Study setting

7. Add a line describing how many centres are involved, and that a description of these centres will follow in future studies. Are the centres in urban or rural areas?

For this study we are working at St Andrews Community church in Waterloo. Added to the section on study setting: "When ran in person, the workshops are run in Central London locations, with a single

centre running the programme in each cycle“. Note from the lead artist: Nationally we are working in Reading, Buckinghamshire, Bristol and Donegal - in both rural and urban centres (outside the study).

Sample and Recruitment

8. Will all participants completing the programme be offered ambassador training, or are they 'hand-picked'? Please provide additional information on the recruitment process, including exclusion and inclusion criteria.

All participants are offered the further training to be ambassadors; the same inclusion/exclusion criteria applies as they are initially recruited as participants for the workshops.

The recruitment process has been described in the recruitment procedure sub-section: "Potential stroke survivor participants will be identified through signposting in community centres and care homes as well as engaging in presentations, screenings, taster sessions and performances during the tour.

Recruitment of potential participants will be done online. Screenings of performance extracts will be followed by taster sessions online and a Q&A with ambassadors. Potential participants will be directed to the project manager at Rosetta Life who will manage all referrals.

Potential participants will be offered a PIS and an ICF and will be explained the details of the study. Written consent will be sought following a 48h colling-off period.

Wider stakeholders will be recruited from the networks of people involved in the referral, delivery or support of the programme.

A recruitment log will be kept by the research team to accurately record included and excluded participants as well as missing data from dropouts to account for possible sampling bias."

Inclusion and exclusion criteria for each group (participants and stakeholders) is described in the section called "sample and recruitment":

"Stroke survivors

Consenting stroke participants will be included if they are:

- (1) over 18 years of age
- (2) have had one or more stroke(s)
- (3) received inpatient care in a UK stroke care pathway
- (4) able to follow a 2-stage command and hold a conversation in English if no supporter/friend is available to translate

The following exclusion criteria will be applied to individuals:

- (1) with co-morbidities that would prevent participation in group activities (e.g. dementia or deteriorating or fluctuating palliative conditions)
- (2) unable to understand English
- (3) unable to commit to the 12-week programme

Additionally, stroke ambassadors will be included if they have been through the ambassador training and are involved in at least one programme cycle culminating in the tour.

All participants will be offered the option of completing an interview with their carer present. This will be offered both after the first 12-week programme and after the ambassador training, for those that wish to participate. Those that decline will be asked if they would be willing to provide their reasons why.

Wider stakeholder group

In addition to the stroke survivors that enrol on the SO programme, data will also be collected from a wider stakeholder group involved in the delivery or support of the programme. Individuals will be recruited if they meet the following criteria:

- (1) over 18 years of age
- (2) can hold a conversation in English if no supporter/friend is available to translate
- (3) can either be defined as a:

- Supporters: family members or carers.
- Deliverers: individuals responsible for the delivery of the research (facilitators and artists).
- Referrers: individuals involved in signposting (e.g. doctors, nurses, healthcare workers).

Wider stakeholders will be excluded from participation if they are unable to understand English or if no supporter/friend is available to translate.”

9. Out of curiosity, what is the aim and purpose of ambassadors? Please clarify the rationale behind having ambassadors and include if they are oriented towards programme participating only or if they are also ambassadors for stroke survivors in other organisations such as hospitals, patient organisations.

Ambassadors advocate for life after stroke offering positive role models how to live a fulfilled independent creative life after stroke, they engage with Rosetta Life programmes and also for stroke survivors in other organisations such as hospitals, patient organisations. From the intervention section: “After the performance is completed, participants will be invited to a four-day training programme where they will learn to act as advocates for life after stroke – termed ‘stroke ambassadors’.”

10. What happens to those who decline ambassador training? Will they too (including wider stakeholders if family members or carers) be subject to assessment and interviews?

Added to the section on sample and recruitment: “Additionally, stroke ambassadors will be included if they have been through the ambassador training and involved in at least one programme cycle culminating in the tour. All participants will be offered the option of completing an interview with their carer present. This will be offered both after the first 12-week programme and after the ambassador training, for those that wish to participate. Those that decline will be asked if they would be willing to provide their reasons why.”

Sampling

Size of sample

11. Rationale for the 75 and 47 included. Is it for power-purposes of the statistical instrument? Please provide a power calculation for this.

Justification of the number of participants: “A prediction of 75 participants have been estimated based on the numbers that over the years running Stroke Odysseys, Rosetta Life has been able to recruit in two consecutive cycles. This number has also considered the organisation being able to whilst maintain a manageable ratio of participants to artists and staff members, guaranteeing that Stroke Odysseys is delivered to the highest standard.”

Justification of the number of stakeholders: “A total of 47 stakeholders, a forecast based on the existing network numbers, will be recruited (12 carers, 10 clinical team members, 5 artists, 20 existing ambassadors).”

12. Power is obviously not needed for studies of qualitative design, but please provide estimation of the number of interviews needed.

At least 20 participants will be interviewed, as well as a subsample (10 in total) of each stakeholder group.

Recruitment procedure

13. Describe who will be managing the recruitment process, that is, describe who keeps track of the recruitment log for you to present an elegant inclusion flowchart in future papers, accounting for possible sampling bias.

Added to the recruitment section: “A recruitment log will be kept by the research team to accurately record included and excluded participants as well as missing data from dropouts to account for possible sampling bias. “

14. As this study is an intervention study with ethical approval, describe the whereabouts of informed and written consent. Potential stroke survivors will see advertisements through signposting and be approached during tours. Will there be information also about the study and not only about the programme? Those who do not wish to participate, are they still enrolled in the programme? To clarify, I am not asking for programme adherence which is well described as an objective in Table 1, but for compliance to study participation.

Added to the recruitment procedure section: "Potential participants will be offered a PIS and an ICF and will be explained the details of the study. Written consent will be sought from following a 48h colling-off period". Rosetta Life runs other Stroke Odysseys programs outside of the SHAPER-SO research and any individual that does not wish to be part of the research will have the opportunity to join a non-research programme.

Data collection

Outcomes

15. Describe who does the assessments and who will be conducting interviews

The Implementation science research team will be conducting interviews with the participants. We reconsidered the applicability of the OX-PAQ clinical measure this is collected in an interview format and, on reflection, the team decided to withdraw this measure from the study as it is not suitable for the wide range of stroke survivors who will take part in the study and the current COVID-19 pandemic (and associated changes in social isolation guidance). This has been clarified in the "semi-structured interviews" section of the manuscript.

16. Describe the process around data storage

The following section has been added:"

Data Protection

The Investigator will ensure that this study is conducted in full conformity with relevant regulations and with the ICH Guidelines for Good Clinical Practice (CPMP/ICH/135/95) July 1996. The Investigator will ensure that this study is conducted by the principles of the Declaration of Helsinki.

Access to person identifiable implementation science data will rest with the data custodian(s) from the immediate study team and the implementation science team. Since the project seeks to explore in some depth participants' experiences and barriers and facilitators to implementation, it is important to maintain strict confidentiality and facilitate openness in the interviews and survey responses thus optimal data quality.

Consent forms and audio/video recordings will be kept electronically in KCL's SharePoint for the duration of the study, only accessible by the teams at KCL, Kingston University and Rosetta Life involved in the study. Consent forms and other identifiable paperwork will be kept in locked cabinets only accessible to the study team. Study data will be kept in a separate location from the personally identifiable information. Access to the de-identified research data will be shared with the study management group for review, analysis and dissemination. Only de-identified data will be analysed. After the completion of the study, the study data will be kept for the King's College London's standard retention period of 10 years. The study data that supports published results will be deposited in a secure data repository (e.g. King's Research Data Management System). This will allow the data to be accessible for future reuse as per King's College London's policy on the management of research data long-term."

Assessment descriptions (page 13)

17. Ethnographic research (line 20): I read '12 workshops' and get a bit confused.

Previously you mention a 12-week programme with workshops, are they 12 workshops 1 week each? If so, please include a table shortly describing the 12 workshops.

The 12-week programme (Performance, Fig 1) is a programme of 12 weekly workshops. Added to the intervention section: "The workshops are the result of co-creation; the general framework is: weeks 1 -

3 building the performance company, weeks 4 - 6 devising the performance weeks 6 - 9 rehearsing the performance and weeks 10 - 12 are sometimes concertina-ed into one production week introducing stage management, lighting and technical runs. Each of the 12 workshops contains a performance "class" of 20 - 30mins exploring movement and voice techniques and exercises."

18. Quantitative assessments (line 58-60): 'investigators will quantify and cost the resources used' – it might be my level of English understanding, but I do not comprehend what they will do. Will the evaluation of the costs (direct and associated) be done about changes in QoL as measured by the Eq5D?

Thank you. We have now made it clearer in the text that costs will be compared to QoL outcomes to inform an analysis of the cost-effectiveness of delivering programmes at scale.

We have added to the Assessment descriptions for the implementation outcomes section: "The EQ5D-3L is a self-complete multi-attribute measure of health-related quality of life that assigns individuals a unique state of health based on their response to individual items."

19. Provide a short description of the Eq5D, including how you plan to deal with the visual analogue scale, EQ VAS gave that some assessment may be done online.

We have provided a brief description of the EQ5D. In this study. We will not be making use of the values from the accompanying VAS. "Utilities" applicable to measured health states using the EQ5D will be derived from pre-existing health state value systems published for the UK population.

We have added to the Assessment descriptions for the implementation outcomes section: "Each unique health state is associated with a pre-determined "utility" value derived from a survey of wider community preferences over different states of health. The utility-scale is anchored at 1 (full health) and zero (death), with negative values allowed in instances where states of health are considered worse than death. Health state utility values are subsequently used to estimate quality-adjusted years survived over time (QALYs) – the utility scores providing the means of making the quality adjustments. Evidence on costs and QALYs will subsequently be used to inform an analysis of the cost-effectiveness of programme delivery at scale. "

Qualitative data collection (page 14)

20. Lines 14-15: clarify whether you refer to the sub-sample when you mention '10 in total' or to a number of wider stakeholder groups. If the latter, it ought to be outlined elsewhere in the method section.

Are we referring to 10 individuals from the wider stakeholder group (for which a total of 47 stakeholders will be recruited - divided into subgroups such as 12 carers, 10 clinical team members, 5 artists, 20 existing ambassadors)? Reworded for clarity: "These issues will also be explored with a sub-sample of individuals (10 in total) from each of the wider stakeholder groups."

Also, please provide information about the following:

21. Demographic and clinical data included. If not included, please provide a rationale for that.

Thank you for your question. Included: "Demographic data will be collected by Rosetta Life at the time of enrolment.

22. Number of groups are needed to reach patient recruitment

Two cycles of the programme will be delivered to reach the patient recruitment target. In the intervention section: "The programme will be delivered in two cycles of the complete 3-stage intervention."

23. Size of the group, including wider stakeholders

Added to the qualitative data collection section: “We will conduct semi-structured interviews with a purposive sub-sample of stroke survivors (N= 20: 5 from each cycle at two-time points – T2 and T3)”.

24. time for data collection, i.e., when does the study start and expected end of data collection (approximate date)

Due to funding body constraints and COVID delays we will refrain from adding a start date and end date of this study. The study has been delayed by 15 months and we are waiting for approval for a costed extension that will allow us to conduct this work fully. The existing start and end dates (start on 01-Sep-2021 and end date 30-Sep-2022) would be misleading for the reader.

Data analysis (page 14)

25. Qualitative analyses (line 42): you will be using an inductive thematic approach. Please clarify ad modum what template. It would provide the reader with enough information about the theoretical framework for the analytical method chosen.

A thematic analysis will be used as the method for identifying, analysing, organizing, describing, and reporting themes found within a data set from participants and stakeholders following transcription of recordings from semi-structured interviews. In the qualitative analysis section: “Initial analysis of qualitative data will be undertaken using an inductive approach to thematic analysis. All data from interviews and observations will be managed using NVivo 10 and examined to categorise themes and key issues that emerge. Using this inductive approach, tentative theoretical explanations will be generated for each sub-group. Summary memos for data sets will be developed for each sub-group to provide the basis for within and between-group comparisons. The inductive approach is data-driven. Based on observation, the early analysis seeks to reveal patterns and themes from which tentative hypotheses can be drawn subsequently leading to theory; theories are devised to explain what is seen rather than the other way around.”

Reviewer #2

1. Please mention - the place of the study in the title.

The place of the study has now been included in the title: “Evaluation of a community-based performance arts programme for people that have experienced a stroke in the UK”.

2. The aim did not include effectiveness however the methods have it. Is this a part of a bigger study if so please revise this section for more clarity. Also include the study design and expected outcomes. These details are lacking in the presentation abstract.

The study design has been included in the abstract: “The study will evaluate the experience and impact of Stroke Odysseys on those participating using mixed methods (interviews, observations and surveys) before and after each stage, and carry out non-participant observations during a percentage of the workshops, training and tour”. Section amended for clarity.

3. The place of study is missing?

This has now been added to the title and the abstract section.

4. If there are any published details of the bigger study - please cite that.

The bigger study (as in the programme that SHAPER-SO is part of) is still ongoing, so there is no published data at present.

5. SHAPER can be spelt out once and the acronym can be used after that.

Amended. SHAPER has been spelt out once in the abstract and once in the main body of the text.

6. The terms evaluation, implementation effectiveness and clinical effectiveness direct the readers thinking towards a trial as opposed to what's mentioned in the methods (in Abstract). This is a mixed-methods study; terminology has been clarified throughout the manuscript.

7. it will be good to add the expected outcomes of SHAPER SO for the UK and the world. The following was added to the introduction section: "Indeed, over the past decade, several studies conducted in this patient population have consistently shown a positive impact of different art modalities on psychological (e.g. enhancement in confidence and a better sense of control), social (e.g. increased social interactions and peer support) and functional (e.g. improvement in physical abilities) outcomes [14]. Nonetheless, despite the growing body of research on the benefits of art interventions, the process of scaling-up these interventions, embedding them into healthcare and its associated challenges are not yet well-established. Preliminary data indicates that Stroke Odysseys (discussed below) is received positively by those who take part, however, identifying barriers to implementation and exploring ways to overcome these obstacles is essential to successfully and sustainably embed SO into clinical pathways and roll out the programme at a wider scale."

8. please over the Aim up and present the objectives after the Aim. Also include the context within the AIM as well as objectives to make it more specific. Also please avoid repetition of the objectives.
Objectives have been presented after the aim and have been streamlined to avoid repetition.

9. It will be good to sub-group these long objectives listed under the objectives mentioned above to relate to the objectives and AIM mentioned above. Are these sub-objectives?
Objectives have been streamlined for clarity.

10. The theoretical underpinning can be more aligned with the objectives of the study. Currently, it is a very useful section but unable to provide clarity on which framework will be used to understand which objective/Sub-objective. Can be improved. The simplest way to do this will be to specify the objectives within the brackets wherever required.
Expanded COM-B in the section on theoretical underpinning: "Moreover, we reviewed the COM-B tool [19] to help us identify any barriers that may affect individual's engagement with the SO programme (objective 7 and 10). COM-B is a behaviour system network that involves three key components: capability, opportunity, and motivation. COM-B components lie at the centre of the Behaviour Change Wheel (BCW), a framework for designing and characterising behaviour change interventions [19]."

11. Please expand COM-B
Expanded COM-B: "Moreover, we reviewed the Capability, Opportunity, and Motivation Model of Behaviour (COM-B) tool".

12. Please mention this in the abstract too.
This has now been added to the abstract.

13. It will be good to add details about how this intervention was developed or cite the paper detailing it.
Added to the intervention section: "Stroke Odysseys is a post stroke performance arts intervention designed and delivered by arts organisation Rosetta Life. The Stroke Odysseys intervention, initially developed and funded by King's and Guy's & St Thomas' Charity, aims to improve recovery, agency and well-being after stroke [25]."

14. Each kind of approach is expected to have a different impact, effectiveness and implementation challenges to document and report.

Thank you for raising these considerations, they will be reported in the findings in a later data paper.

15. Are there any insurance cover for these participants? especially those who want to attend it in person?

Yes, King's College London holds negligent harm and non-negligent harm insurance policies which apply to this study.

16. What kind of support will be provided to those severely disabled but wish to attend face to face or even otherwise. Especially stroke has its impact on the cognitive capacities of the individual. How will such survivors understand, engage and learn? Please describe.

Rosetta Life have successfully included people with severe cognition damage in previous programmes in the past. With the support of carers and staff members, it has been possible to include people who were severely disabled. For this study, they will be included if they meet the inclusion criteria.

17. It is assumed that the readers will understand that the study is taking place in the UK. However, they must explain where exactly this is happening.

Due to the changing nature of COVID restrictions, we are unable to pinpoint where the intervention will take place from the outset. We predict that in-person sessions will be delivered around central London and Oxfordshire but for the online approach, participants might experience the intervention from anywhere in the UK. For this, we have kept the location as the UK.

18. The authors could present the methods in PICO format having participants first described followed by the intervention.

The authors are unsure what is being requested here. The PICOT process would result in the following statement

Stroke survivors (P), Stroke Odysseys (I),? (Compare), Evaluation (Outcome),? (Time / Type of Study or Question). This may not be the best format to present the question in a mixed-methods study that is not strictly an RCT on CTIMPs.

19. The experiences differ for the first stroke survivor and those with recurrent stroke - the authors need to describe how they will address these differences.

This is a very poignant argument. However, this programme is inclusive of all stroke stories, from recent to historical and so dissecting these differences is not the main aim of the research. We will be looking at issues around implementation; first stroke/recurrent stroke/historical stroke could all have very different needs (even those who have just had one stroke -i.e. some will be affected more than others). In the interviews, we will ask them when they had their stroke, how it has impacted their life etc.

20. will the authors assess it using a standardised scale? like NIH?

The two-stage command will be assessed in situ with the participant, verbally, with researchers that have vast experience working with the stroke survivor population. The NIH scale, used frequently in clinical settings, was deemed to be burdensome in addition to all the other measures that are being collected at the initial stages from this population.

21. Those who have COVID-19 experience and present with the long covid condition?

We will apply the following criteria: "co-morbidities that would prevent participation in group activities", if long COVID prevents participants from participating, they will not enrol/withdraw as per their decision; we will not exclude participants due to long COVID.

22. the authors need to justify very clearly how did they arrive at this number? It also needs to be very clearly reported that having 3 objectives and 10 subjective - how do the authors think the study will be sufficiently powered both qualitatively and quantitatively to achieve the objectives.

The sample in this mixed-methods study was calculated based on the number of people that Rosetta Life, over the years, knows they can recruit over two cycles. Due to the nature of this mixed-methods study, which heavily relies on qualitative measures, a power calculation was not deemed necessary.

23. Here comes the importance of study sites. How many sites and in which regions. Especially because one hospital or an area may have 100 stroke survivors but very few are willing to consent. Especially given the COVID-19 situation despite their stroke severity.

For this study we are working at St Andrews Community church in Waterloo. Added to the section on study setting: "When ran in person, the workshops are run in Central London locations, with a single centre running the programme in each cycle". Note from the lead artist: Nationally we are working in Reading, Buckinghamshire, Bristol and Donegal - in both rural and urban centres (outside the study).

24. These need to be very clearly described. I did not come across any quantity or quali design so far in the manuscript that relates to the objectives with a specified group of participants. FOr example - How, where and why semi-structured interviews will be conducted. What will be the use of that data?

Semi-structured interviews will be conducted for qualitative data after the performance (12-week workshops) and quantitative data will be collected after ambassador training, The data will be used to address the aim and objectives presented above.

25. This is helpful but the authors need to provide a rationale for using a quantitative method to assess a qualitative aspect that could provide great insights when assessed qualitatively. This is another example that I can refer to in my previous comment. I may not like something but if you ask me to complete a questionnaire - I will probably say I liked it.

Thank you for your comment. We will not be using a quantitative method to assess a qualitative aspect; interviews will be used to explore views on the programme. We reviewed and clarified the measures and methods of data collection.

26. Could be better if the authors can develop a similar table for the objectives or cite this table right at the beginning. Also, include the sample size for each of the objectives.

Thank you for your suggestion. Please see tables 1 and 2. Table 1 lays out the clinical objectives, outcomes, types of assessment and timepoint for data collection. Table 2 lays out the implementation objectives, outcomes, types of id assessment, time points and who the data is being collected from.

27. This section can go to the supplementary file if there is a need to reduce the word count. There is also a need to include - who will collect the data as the answers provided depends on who collects it.

The researchers collecting each type of data has now been added to this section.

28. Although detailed., I will say that this section is very generic. The authors must certainly provide more details on which kind of analysis will be used to achieve which objectives specifically. How data will be triangulated if the same methods and analysis are used for many objectives etc. Further details on the analyses have been added to this section.

29. I still did not come across the section on Expected outcomes, expected plans for scalability the implementation - what if the results go in another direction and what will be implications. Authors could add these details.

This is the first scalability study on Stroke Odysseys and so there are no defined expected outcomes based on previous evaluations of this intervention – since these have not been conducted yet. The plans for scalability are to be defined once the data from this study inform scalability implications from an implementation and health economics points of view.

30. Please describe the duration difference between the baseline/midpoint and end, line evaluation.

The differences in time points have been listed below:

Baseline (before the start of the 12-week workshops) – T0

Performance (12-week workshops) – T1 (around week 6), T2 (week 12 onward)

Tour (ongoing after T2)

Stroke Ambassador Training (4 weeks workshops), T3 (after 4 weeks ambassador training)

VERSION 2 – REVIEW

REVIEWER	Markovic, Gabriela Karolinska Institute
REVIEW RETURNED	01-Feb-2022
GENERAL COMMENTS	I am very pleased over the revised manuscript, it is clear and describes the process in a elegant and transparent way. Looking forward reading about the outcomes!